# GenWarp: Single Image to Novel Views with Semantic-Preserving Generative Warping

**Junyoung Seo**[1,3] *   **Kazumi Fukuda**[1]   **Takashi Shibuya**[1]   **Takuya Narihira**[1]   **Naoki Murata**[1]
**Shoukang Hu**[1]   **Chieh-Hsin Lai**[1]   **Seungryong Kim**[3†]   **Yuki Mitsufuji**[1,2†]
[1]Sony AI        [2]Sony Group Corporation        [3]KAIST AI

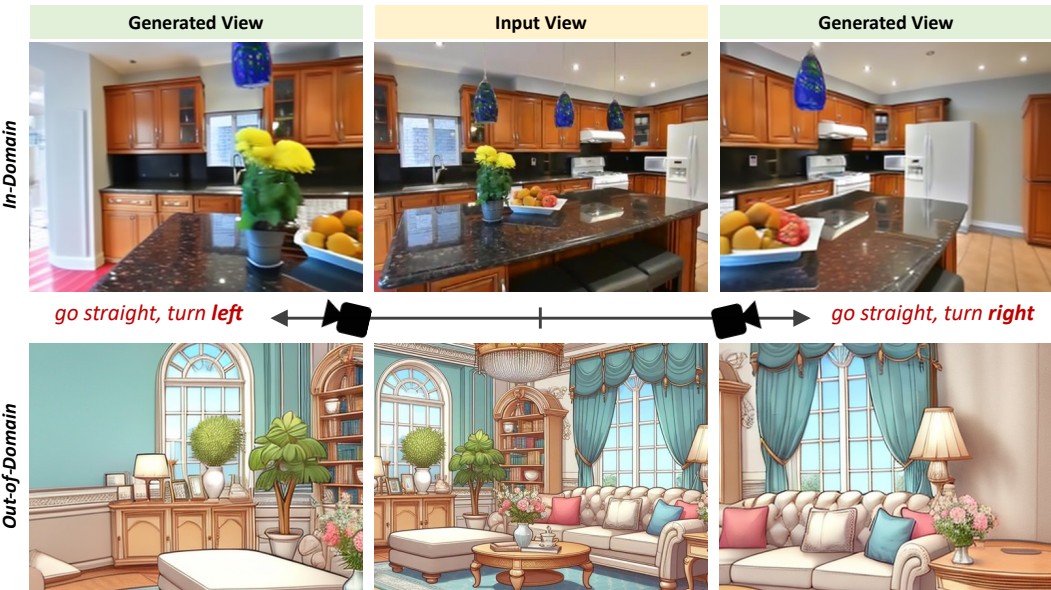

Figure 1: **Teaser.** Our model generates plausible novel views, conditioned on **only a single input view**, enabling to handle both in-domain images (top) and out-of-domain images (bottom).

## Abstract

Generating novel views from a single image remains a challenging task due to the complexity of 3D scenes and the limited diversity in the existing multi-view datasets to train a model on. Recent research combining large-scale text-to-image (T2I) models with monocular depth estimation (MDE) has shown promise in handling in-the-wild images. In these methods, an input view is geometrically warped to novel views with estimated depth maps, then the warped image is inpainted by T2I models. However, they struggle with noisy depth maps and loss of semantic details when warping an input view to novel viewpoints. In this paper, we propose a novel approach for single-shot novel view synthesis, a semantic-preserving generative warping framework that enables T2I generative models to learn *where to warp* and *where to generate*, through augmenting cross-view attention with self-attention. Our approach addresses the limitations of existing methods by conditioning the generative model on source view images and incorporating geometric warping signals. Qualitative and quantitative evaluations demonstrate that our model outperforms existing methods in both in-domain and out-of-domain scenarios. Project page is available at `https://GenWarp-NVS.github.io`.

---

*Work done during an internship at Sony AI.    † Co-corresponding authors.

38th Conference on Neural Information Processing Systems (NeurIPS 2024).

# 1 Introduction

Text-to-image (T2I) diffusion models (*e.g.*, Stable Diffusion [35]) have made rapid progress in generating diverse high-quality images when given a user text prompt. This holds extensive potential utility across various domains, including portrait photo design, cartoon creation, and movie production. However, current T2I models lack the flexibility of moving cameras in the generated image. For example, when a user tries to move the camera closer or farther to the generated image, T2I models often fail to change the viewpoint of the generated image with a proper notion of 3D awareness. This limits its application in real-world scenarios where we hope to achieve user-tailored designing purposes by changing the camera viewpoint for generated images.

To freely move camera viewpoints of an image, a line of research focuses on directly learning a single-shot novel view generation model with a camera viewpoint condition from large-scale 3D datasets. For example, with the advent of large-scale 3D object datasets such as Objaverse [10], recent attempts [26, 40, 27] achieve success in generating novel views of 3D objects from a single image. Beyond the object-centric novel views, efforts for full 3D scenes have also been made [34, 22, 36, 17, 7]. Unlike the object-centric models, these can generate novel views of complex scenes from a single image. The performance of single-shot 3D scene novel view generation models highly depends on the scale of multi-view 3D scene datasets [52, 8, 5]. Compared with the object-centric multi-view datasets [10, 9], it is hard to collect such a large-scale dataset for 3D scenes due to its complexity. Thus, existing models [34, 22, 36, 17, 7] solely trained on these datasets [52, 8, 5] struggle to handle in-the-wild images [17, 36].

Instead of learning dataset-specific novel view synthesis models, alternative approaches [7, 31] propose utilizing the generative prior from large-scale T2I diffusion models, *e.g.*, Stable Diffusion [35]. These works adopt a two-step strategy for novel view generation, called *warping-and-inpainting*, similarly to conventional works [48, 34, 22], with a combination of the large-scale T2I diffusion models and off-the-shelf monocular depth estimation (MDE) models (*e.g.*, MiDaS [33], ZoeDepth [2]). Specifically, they first predict a depth map of a given image via off-the-shelf MDE models [2, 33], and then warp the input image to novel camera viewpoints with the depth-based correspondence, followed by inpainting occluded regions of the warped images with proper text prompts through the T2I diffusion models. The warping-and-inpainting approach successfully generates novel views from in-the-wild images by utilizing large capabilities of T2I diffusion models learned from large-scale image datasets [39].

Despite such an advantage, this warping-and-inpainting approach can generate novel views only in a limited range of camera viewpoints around the input image. This is because **(1) they struggle to handle noisy depth maps predicted by the MDE**. As shown in Fig. 2(a), a reprojection error from the estimated depth map makes the warped image unreliable, becoming a significant performance bottleneck. The subsequent inpainting T2I diffusion models cannot refine the artifacts caused by this error. In addition, **(2) important semantic details of the input view sometimes get lost during geometric warping**, especially when dealing with challenging camera viewpoints. In the above two cases, only a sparse set of pixels is preserved in the warped image, making it difficult to generate the occluded regions while preserving the semantic information of the input view. Fig. 2(b) shows a clear example of this problem; the inpainted regions show a different context with the input view.

To address these issues, we propose a generative warping framework, **GenWarp**, in which we make T2I generative models learn *where to warp* and *where to generate* in images, instead of inpainting unreliable warped images. Our generative model integrates view warping and occlusion inpainting into a unified process, unlike existing two-step approaches that perform these operations separately. By directly taking the input view images with their estimated depth maps, our model learns to warp them and to generate the occluded or ill-warped parts with our augmented self-attention. Our approach eliminates the dependency on unreliable warped images and integrates semantic features from the source view, preserving the semantic details of the source view during generation. Similar to [26, 40, 49], we leverage generalization capabilities of T2I diffusion models (*e.g.*, Stable Diffusion [35]) through fine-tuning a T2I diffusion model.

Our main contributions are as follows:

- We propose a semantic-preserving generative warping framework, **GenWarp**, to generate high-quality novel views from a single image.

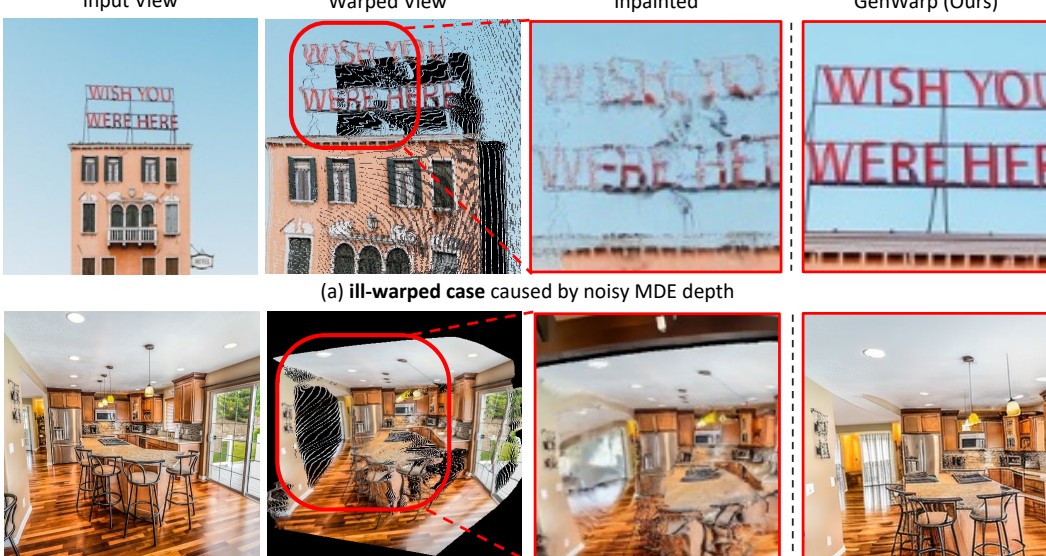

|  Input View | Warped View | Inpainted | GenWarp (Ours) |

(a) **ill-warped case** caused by noisy MDE depth

(b) **missing original semantics** of severely occluded regions

Figure 2: **Limitations of explicit warping-and-inpainting approach [35, 7, 31].** Results from challenging new camera viewpoints for warping-and-inpainting approach show artifacts. (a) The neon sign present in the input view is distorted after geometric warping due to the noisy depth. (b) The next room peeked in from the new camera viewpoint lacks the context given by the input view.

- **GenWarp** learns *where to warp* and *where to generate* in images through augmenting self-attention with cross-view attention instead of inpainting unreliable warped images, which eliminates the artifacts caused by error depths and integrates semantic features from source views, preserving semantic details in generation.

- Extensive experiments on RealEstate10K [52], ScanNet [8], and in-the-wild images (*e.g.*, AI-generated images) validate that **GenWarp** achieves superior performances over existing methods in both in-domain and out-of-domain scenarios.

## 2 Related Work

Generating novel views from a single image is a challenging ill-posed problem that has primarily been addressed in combination with generative modeling. These novel view generative models can generally be categorized into two types: those designed for object-centric scenes and those designed for general scenes, including indoor and outdoor scenes. On the other hand, following the recent success of large-scale T2I models, there are methods that can control the generation results, exploiting the attention mechanism.

**Single-shot novel view synthesis for objects.** Following the success of image diffusion models [15, 35], diffusion models for novel view synthesis [47, 4] have been proposed. These works train diffusion models to take a single image and a novel camera viewpoint as conditions, and directly generate novel view images. More recently, with the emergence of large-scale 3D datasets such as Objaverse [10, 9], generalized generative models for single-shot novel view synthesis (NVS) have emerged. Recent works [26, 27, 19], including Zero123 [26], achieves powerful generalization capability by fine-tuning T2I diffusion models on Objaverse. While these models enable object-centric novel view synthesis from an in-the-wild single image, such generalized novel view models for general scenes remain relatively unexplored.

**Single-shot novel view synthesis for general scene.** Single-shot NVS often necessitates generating outer regions or occluded regions that are not visible in an input view. Thus, recent works [48, 34, 25, 22] propose generating novel views in a warping-and-refining fashion, which involves first predicting a depth map of an input view, then warping the input view along with the depth map to a desired viewpoint, and finally refining missing regions arising from the geometric warping. Another line of works [36, 44, 17] directly train novel view generative models without depth-based warping. For

example, GeoGPT [36] achieves novel view generation, by feed-forwarding an input view and a camera viewpoint to a transformer-based architecture. Other recent works [44, 17] train novel view diffusion models with a cross-view attentions [44, 17], or an epipolar constraint [44]. More recently, some works [31, 7, 41] have proposed bringing a large-scale T2I model [35] to the warping-and-refining strategy. This approach enables the generation of novel views from in-the-wild images, which was previously challenging. Nonetheless, it shows unstable results, especially when the camera viewpoint is far from its original position. Concurrently, ZeroNVS [38] fine-tunes Zero123 [26] for NVS in general scenes. It focuses on camera parametrization to avoid 3D scale ambiguity, which differs from our focus; we focus on improving depth warping-based NVS.

**Attention-based control in large-scale T2I models.** Since the emergence of large-scale text-to-image (T2I) diffusion models [35, 37], recent works [21, 40, 3, 14, 16] have investigated the properties of self-attention within T2I models. For example, Text2Video-Zero [21] and MVDream [40] generate consistent images by sharing self-attention between video frames or 3D multi-views, respectively. Similarly, Animate-Anyone [16] and MagicAnimate [49] generate human dance videos through a fine-tuned T2I model that shares self-attention with input image features. Observing the generalization capability and efficiency of these self-attention-based controllable architectures, our approach is highly influenced by them. However, using these architectures for single-shot novel view generation is non-trivial, as input scenes are usually complex, and the details within them must be generated consistently with the camera movements. Our model integrates MDE depth-based correspondence while benefiting from the advantages of these architectures, thereby significantly improving single-shot NVS performance.

## 3 Method

### 3.1 Preliminaries and problem statement

Given a single image for an input view $I_i$, our goal is to generate a novel view $I_j$ from a relative camera viewpoint $P_{i \to j}$ and a camera intrinsic $K$. To enable this, recent works [7, 31] propose a *warping-and-inpainting* framework that adopts monocular depth estimation (MDE) models [2] for geometric warping and T2I generative models [35, 37] for inpainting. In this approach, an input view image $I_i$ is geometrically warped with its MDE depth map $D_i$ to the desired camera viewpoint $P_{i \to j}$:

$$I_{\text{warp}} = \text{warp}\big(I_i; D_i, P_{i \to j}, K\big), \tag{1}$$

where $\text{warp}(\cdot)$ is a geometric warping function which unprojects pixels of an input view image $I_i$ with its depth map $D_i$ to 3D space, and reprojects them based on the desired camera conditions, $P_{i \to j}$ and $K$. More specifically, in homogeneous coordinates, pixel location $x_i$ in the input view is transformed to the pixel location $x_j$ in the novel view such that:

$$x_j \simeq K P_{i \to j} D_i(x_i) K^{-1} x_i, \tag{2}$$

where the projected coordinate $x_j$ is a continuous value. To obtain the warped image $I_{\text{warp}}$, it is followed by mapping the pixel colors from the input view to the novel view with a flow field between $x_i$ and $x_j$ [48, 30, 34]. And, inpainting diffusion model $\phi$ generates a novel view image by filling the missing regions in the warped image $I_{\text{warp}}$:

$$I_j \sim p_\phi(I_j; I_{\text{warp}}, M_{\text{warp}}, c_j), \tag{3}$$

where $p_\phi(\cdot)$ is a learned distribution of the diffusion model $\phi$ conditioned on an occlusion mask $M_{\text{warp}}$, a text prompt $c_j$, and the warped image $I_{\text{warp}}$ used for inpainting. This approach assumes that the estimated depth map $D_i$ is accurate. Under the assumption, the warped image $I_{\text{warp}}$ would be an ideal guidance for generation.

However, we have observed that the warped image is often not reliable when a novel camera viewpoint is far from the original viewpoint. It is because this explicit warping operation is sensitive to errors in the depth map, and depth maps predicted by MDE are usually noisy, arising artifacts after the warping. The subsequent inpainting model only takes the warped image $I_{\text{warp}}$ as input which contains ill-warped artifacts outside the region to be inpainted, thus showing limited performance at large view changes. Additionally, the warped image may lose the semantic information originally contained in the input view due to factors such as occlusion, but this approach does not take that into account, as exemplified in Fig. 2.

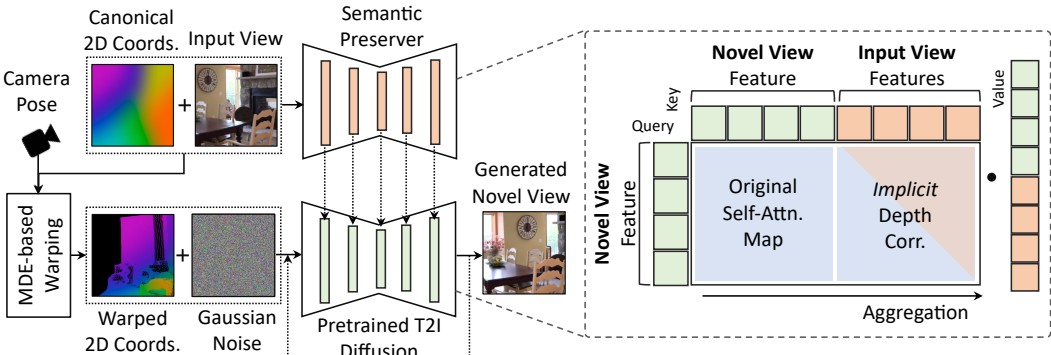

Figure 3: **Method overview:** (**Left**) Given an input view and a desired camera viewpoint, we obtain a pair of embeddings: a 2D coordinate embedding for the input view, and a *warped* coordinate embedding for the novel view from estimated depth through MDE. With these embeddings, a semantic preserver network produces a semantic feature of the input view, and a diffusion model conditioned on them learns to conduct geometric warping to generate novel views. (**Right**) We augment self-attention with cross-view attention, followed by aggregating the features with both attentions at once. It helps the model to consider where to generate and where to warp.

## 3.2 Semantic-preserving generative warping

To alleviate the aforementioned limitations, we introduce a novel approach where a diffusion model learns to implicitly conduct geometric warping operation, instead of warping the pixels or the features directly. We design the model to interactively compensate for the ill-warped regions during its generation process, thereby preventing artifacts typically caused by explicit warping. In addition, to preserve the semantics in the input view, our framework takes the input view image without warping, and the encoded semantic features of the input view are incorporated into the generation process, which is different from other approaches that solely take an unreliable warped image from which the original semantics are difficult to infer.

To this end, we leverage the attention layers inside the pre-trained diffusion U-net [35]. Our key idea is to learn the attention between input view and novel view features, which serves as an *implicit* correspondence that mimics explicit depth-based warping within the diffusion model. By incorporating this into the diffusion process, we aim to seamlessly integrate the effect of depth-based warping into the generative prior. This implicit correspondence in the form of attention can be integrated into the existing self-attention layers inside the diffusion U-net. In so doing, the input view features additionally interact with the novel view features in the generation process, making the diffusion models naturally find *where to generate* and *where to warp*, as visualized in Fig. 4.

**Two-stream architecture.** Our approach comprises a two-stream architecture, a semantic preserver network and a diffusion model, sharing an identical U-net-based architecture. The semantic preserver network takes the input view image $I_i$ and produces a semantic feature $F_i$ of the input view. And, the diffusion model generates a novel view image $I_j$, by integrating the input view feature $F_i$ into its internal novel view feature $F_j$. To imbue the diffusion model with the MDE depth-based correspondence, we use a pair of canonical coordinates and warped coordinates as additional conditions. Fig. 3 illustrates an overview of our architecture. In the following, we explain each component in detail.

**Warped coordinate embedding.** To condition on the MDE depth-based correspondence, we use two coordinate embeddings, a canonical coordinate embedding for the input view, and a *warped* coordinate embedding for the novel view. We are motivated to use the warped coordinate embedding by [29], whose purpose is correspondence-based appearance manipulation. Here, we extend this concept to the geometric warping for novel view generation.

Specifically, we construct a canonical 2D coordinate map $X \in \mathbb{R}^{h \times w \times 2}$, where each value is normalized between $-1$ and $1$. This 2D coordinate map is transformed by a positional encoding function $\gamma$ into Fourier features [43] $C_i = \gamma(X)$. We use this Fourier feature map $C_i$ as the coordinate embedding for the input view $I_i$. We then geometrically warp this coordinate embedding $C_i$ of the

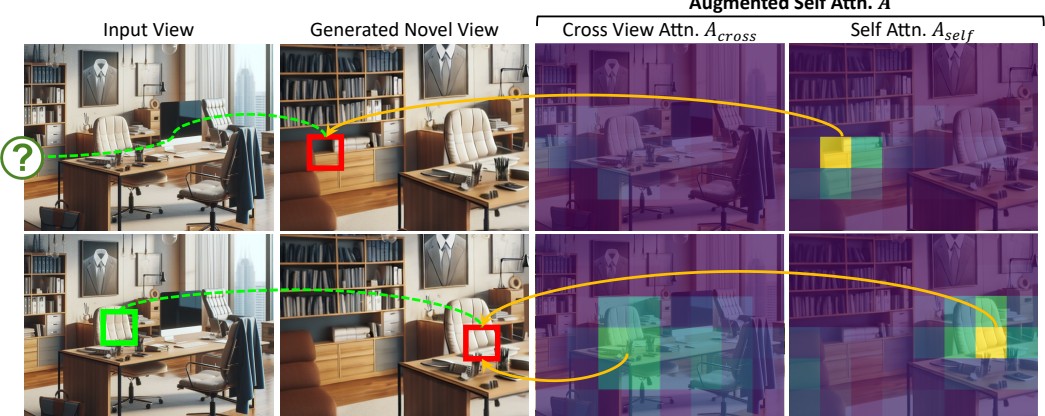

Figure 4: **Visualization of augmented self-attention map.** In augmented self-attention map $A$, the original self-attention part $A_{\text{self}}$ is more attentive to regions requiring generative priors, such as occluded or ill-warped areas **(top)**, while the concatenated cross-view attention part $A_{\text{cross}}$ focuses on regions that can be reliably warped from the input view **(bottom)**. By aggregating both attentions at once, the model naturally determines which regions to generate and which to warp.

input view to the desired novel viewpoint $P_{i \rightarrow j}$:

$$C_j = \text{warp}(C_i; D_i, P_{i \rightarrow j}, K), \tag{4}$$

where $\text{warp}(\cdot)$ is the same geometric warping function in Eq. 1. The warped coordinate embedding $C_{\text{j}}$ serves as the coordinate embedding for the novel view $I_j$. These coordinate embedding $C_i$ for the input view and $C_j$ for the novel view are added to the source view feature $F_i$ and the target view feature $F_j$ through convolution layers respectively. This embedding strategy guides the model to follow the geometric correlation between the input view and the novel view. In an explicit warping strategy, the process is inherently affected by depth estimation errors in virtue of regarding the warped image as a reliably given condition. However, by implicitly learning to warp with this embedding, it is expected that the influence of these errors can be mitigated.

**Augmenting self-attention with cross-view attention.** To infuse the input view features $F_i$, we first construct a cross-view attention, where the cross-view attention map represents the similarities between the input view and the novel view being generated. Thanks to the coordinate embeddings, the cross-view attention map learns to give depth-based correspondence that can be absorbed into the generative process. Then, we propose concatenating this cross-view attention to the existing self-attention.

Specifically, we concatenate the keys and values of self-attention layers in the diffusion U-net with the input view features $F_i$, and apply the self-attention [45] with the following query, key, and value:

$$q = F_j, \quad k = [F_i, F_j], \quad v = [F_i, F_j], \tag{5}$$

where $F_i$ is the input view feature in the semantic preserver network, and $F_j$ is the novel view feature in the diffusion U-net. Then we can obtain the augmented self-attention map $A$, which is a concatenation of the cross-view attention map $A_{\text{cross}}$ and the self-attention map $A_{\text{self}}$.

By aggregating the values with both attentions at once, the model learns to balance the contributions from the novel view's self-attention $A_{\text{self}}$ and the cross-view attention $A_{\text{cross}}$. Our intuition behind this design is that the original self-attention $A_{\text{self}}$ in the diffusion U-net attends to the generative prior, and the cross-view attention $A_{\text{cross}}$ attends to the warping prior from the input view. This allows the model to inherently decide which regions should rely more on its generative capability and which areas should depend primarily on the information from the input view warping, as shown in Fig. 4.

### 3.3 Training strategy

**Fine-tuning pretrained text-to-image diffusion models.** We leverage the pretrained Stable Diffusion 1.5 model [35] for both diffusion U-net and semantic preserver network, to inherit its generalization capability. Different from Stable Diffusion which takes text prompt embedding through

| Input View | GenWarp (Ours) | SD-Inpainting | Input View | GenWarp (Ours) | SD-Inpainting |

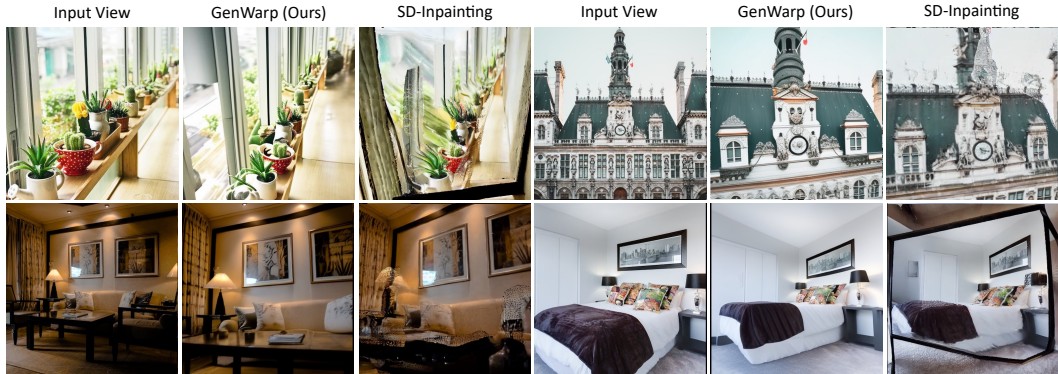

Figure 5: **Qualitative results with images in the wild.** We compare our method with Stable Diffusion Inpainting [35] on in-the-wild images. More qualitatvie results can be found in Fig. 10 of Appendix.

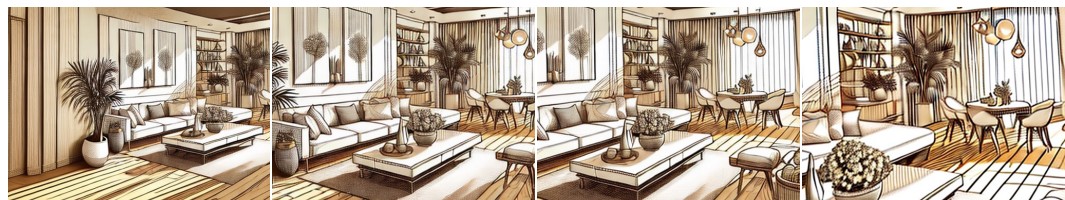

Figure 6: **Consistent view generation results.** Our model can generate consistent multiple views by taking pre-generated novel views as inputs.

CLIP [32], our model takes an image and a desired camera viewpoint as inputs. Therefore, we replace text condition needed for Stable Diffusion with image embedding of the input image through a CLIP image encoder. As all the components in our framework can be trained in an end-to-end manner, we use a sole training loss for fine-tuning, which is the same as the original training loss in LDM [35]. Given a dataset $\mathcal{X}$ consisting of pairs of source view image $I_i$, target view image $I_j$, their camera information $P_{i \to j}$, and a depth map $D_i$, we first encode the source view $I_i$ and the target view image $I_j$ to their corresponding latents $z_i$ and $z_j$ through the LDM encoder, respectively. Then the model is fine-tuned using the following loss function:

$$\mathcal{L}_{\mathrm{ours}}(\theta, \psi) = \mathbb{E}_{\mathcal{X}, t, \epsilon}\left[\|\epsilon - \epsilon_{\theta, \psi}(z_{j,t}; z_i, D_i, P_{i \to j}, K)\|_2^2\right], \tag{6}$$

where $z_{j,t}$ denotes a noised latent of $z_j$ at diffusion timestep $t$. $\epsilon_{\theta, \psi}(\cdot)$ is our model including the diffusion U-net $\theta$ and the semantic preserver network $\psi$, which predicts the added noise in the diffusion process.

**Data preparation.** We fine-tune the model on multi-view datasets including indoor scene and outdoor scene, *i.e.*, RealEstate10K [52], ScanNet [8], ACID [25]. Specifically, we sample two consecutive frames at intervals of 30-120 frames to make pairs of source view and target view images. For ScanNet [8], we use provided ground-truth depth maps and the camera information. For RealEstate10K [52] and ACID [25], ground-truth depth maps are not provided. So, we pre-process the datasets to generate pseudo ground-truth depth maps and their corresponding camera information. Specifically, we use DUSt3R [46] as a pair-depth estimator, followed by PnP-RANSAC [13, 23] to find the corresponding camera information aligned with the estimated depth maps. Additionally, we exclude pairs with low-confident depth maps in our training dataset. Note that our model is less affected by 3D scale ambiguity [6, 38] in this procedure, as camera parameters are aligned to the scales of estimated depth maps.

## 4 Experiments

### 4.1 Experimental setup

We train our model on multiple datasets, including indoor RealEstate10K [52], ScanNet [8], and outdoor ACID [25] datasets. For fair quantitative and qualitative comparison with baseline methods [36, 17] trained on RealEstate10K [52], we also prepared a version of our model trained on the same single dataset. Our baseline methods include GeoGPT [36], Photometric-NVS [17], and the

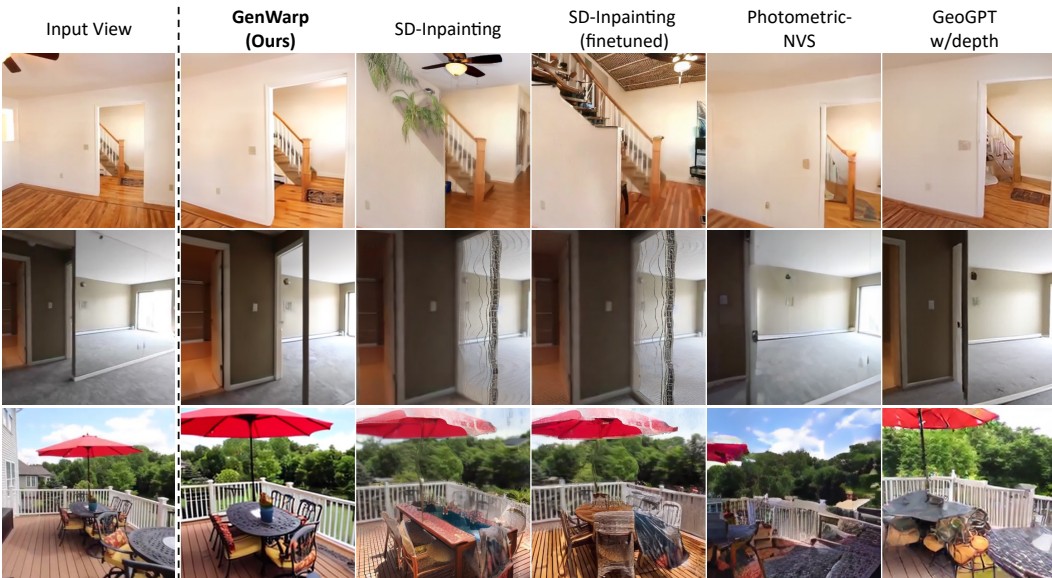

| Input View | GenWarp (Ours) | SD-Inpainting | SD-Inpainting (finetuned) | Photometric-NVS | GeoGPT w/depth |

Figure 7: **Qualitative comparisons with baseline methods [35, 17, 36].** We present single-shot novel view generation results with large viewpoint changes on RealEstate10K [52] test set. Our GenWarp generates high-quality novel views consistent with the input views. We also provide qualitative results on ScanNet [8] in Fig. 9 of Appendix.

warping-and-inpainting method [7, 31] using the Stable Diffusion Inpainting model [35]. To ensure a fair evaluation, we also provide results of the Stable Diffusion Inpainting model fine-tuned on the same multi-view dataset [52]. For GeoGPT, we compare with the results from the depth conditional setting, which is most similar to our approach. For the methods that utilize depth information, namely, our method, GeoGPT, and Stable Diffusion Inpainting, we use the same monocular depth estimation models [2, 46]. Please refer to Appendix B for additional details.

## 4.2 Qualitative results

**Qualitative results on in-the-wild images.** Fig. 5 and Fig. 6 show qualitative results on in-the-wild images, *e.g.*, cartoonish pictures, real photos, and AI-generated [1] images. SD-Inpainting [35]-based warping-and-inpainting approach, used in recent works [7, 31], shows reasonable results with a good generalization capability for extrapolation, but ill-warped artifacts exist in some areas. In contrast, our method consistently generates feasible novel views by refining those artifacts well.

**Qualitative comparisons.** We present qualitative comparisons with the baseline methods [35, 17, 36] in Fig. 7. The warping-and-inpainting approaches with the SD-Inpainting model [35, 7, 31] show good performance for areas where the input view and novel view clearly overlap. However, for regions where warped pixels are sparse, it generates inconsistent novel views without considering the semantic information of the input view. Photometric-NVS [17] and GeoGPT [36] show reasonable performance when the camera view changes are small. However, their performance degrades when the view change is large or when the given images of scenes are underrepresented in the training data, such as outdoor scenes. Our method generates plausible novel views and is robust to variations in the type of scenes and camera viewpoints, by considering the semantics of the input views.

## 4.3 Quantitative results

We perform a quantitative comparison of our model and baseline models [35, 36, 37] trained on RealEstate10K [52], on the test set of RealEstate10K (in-domain) and ScanNet [8] (out-of-domain) using FID for generation quality on distribution level and PSNR for reconstruction quality, with 1,000 generated images. We categorize the distance between source and target views into mid range (30-60 frames) and long range (60-120 frames). Tab. 1 demonstrates that our method shows superior performance in both out-of-domain setting and in-domain setting. The SD-Inpainting-based approaches perform well in terms of PSNR thanks to explicit warping, but struggle with ill-warped artifacts resulting in poor FID. GeoGPT shows good generation quality as evidenced by its FID score

| Methods | Out-of-domain [8] | | In-domain [52] | | | |
| | Mid range | | Mid range | | Long range | |
| | FID ↓ | PSNR ↑ | FID ↓ | PSNR ↑ | FID ↓ | PSNR ↑ |
|---|---|---|---|---|---|---|
| GeoGPT [36] w/depth | 85.52 | 11.36 | 32.70 | 12.26 | 33.91 | 11.69 |
| Photometric-NVS [17] | N/A | N/A | 37.17 | 12.05 | 39.93 | 11.63 |
| SD-Inpainting [35, 7] | 52.20 | 11.68 | 41.76 | 14.21 | 44.13 | 12.98 |
| SD-Inpainting [35, 7] (fine-tuned)[†] | 72.90 | 9.10 | 39.17 | 14.35 | 43.08 | 13.10 |
| GenWarp (Ours) | **46.03** | **12.95** | **31.10** | **14.55** | **32.40** | **13.55** |

Table 1: **Quantitative comparisons.** We compare our method with novel view generative models [36, 17] and warping-and-inpainting approach consisting of Stable Diffusion Inpainting [35, 7, 31], on in-domain setting (training dataset [52]), and out-of-domain setting (external dataset [8]). [†] We additionally provide results of Stable Diffusion Inpainting fine-tuned on the multi-view dataset [52].

but tends to disregard input view details, leading to poor PSNR. In the out-of-domain setting, the SD-Inpainting shows reasonable performance, but its performance deteriorates after fine-tuning on the multi-view dataset [52]. For Photometric-NVS [17], we exclude its out-of-domain results as its provided model trained on RealEstate10K [52] fails to generate novel views when camera parameters in ScanNet [8] are given.

## 4.4 Ablation study

**Embeddings for warping signal.** We perform an ablation study on the embeddings used to create the warping signal by geometric warping with MDE depth maps. We compare the warped coordinate embedding to other possible candidates: warped depth and warped image. Additionally, we test the camera embedding as condition. Specifically, we encode the camera viewpoint to a Plücker ray representation [42] and replace the warped coordinate embedding with this camera embedding. As shown in Tab. 2, the warping signal given by the warped coordinate embedding is the most effective among them.

| Conditions | FID ↓ |
|---|---|
| Warped coordinates | **32.40** |
| Warped depth map | 34.17 |
| Warped image | 35.27 |
| Camera embedding [42] | 39.10 |

Table 2: **Ablation on embeddings.**

**Camera viewpoint variations.** Fig. 8 illustrates the relationship between the difficulty of camera viewpoint changes and the degree of distortion in the generated novel views. We adopt the analysis of view changes proposed in [36], using the LPIPS [51] metric between GT source and target views as a proxy for viewpoint change difficulty, and the LPIPS between the generated and GT target views as a measure of distortion. As shown in Fig. 8, our method achieves the least distortion compared to the baseline methods. Inpainting-based methods [35] show the second-best performance when the viewpoint change is not large, but GeoGPT [36] shows better performance in the case of extreme viewpoint changes.

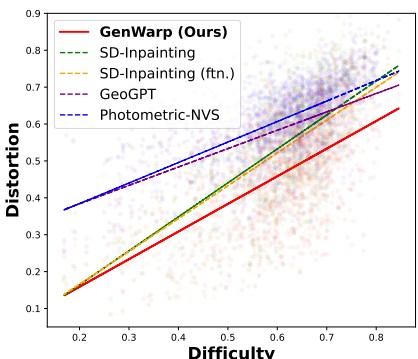

Figure 8: **Comparison on various viewpoint changes.**

## 5 Conclusion

We have proposed **GenWarp**, a framework for generation of novel views from a single image, preserving semantics contained in the input view by learning to warp images through a generative process. By augmenting the self-attention in diffusion models with cross-view attention conditioned on the warping signal, our approach learns to preserve the semantics of the input view while naturally determining where to warp and where to generate. Extensive experiments demonstrate that GenWarp generates higher-quality novel views compared to existing methods, especially for challenging viewpoint changes, while exhibiting generalization capability to out-of-domain images.

## Societal Impacts

This paper presents in the field of AIGC (AI-Generated Content). The proposed model in the paper generates images of user-provided camera viewpoints based on input images. Therefore, while there may be potential social impacts as a consequence, there is nothing in particular to be highlighted. Our model relies on learning from large-scale multi-view datasets, so it may reflect potential societal biases included in these datasets.

## Acknowledgment

This research was supported by Institute of Information & communications Technology Planning & Evaluation (IITP) grant funded by the Korea government (MSIT) (RS-2019-II190075, RS-2024-00509279) and the Culture, Sports, and Tourism R&D Program through the Korea Creative Content Agency grant funded by the Ministry of Culture, Sports and Tourism (RS-2024-00345025, RS-2023-00266509, RS-2024-00333068), and National Research Foundation of Korea (RS-2024-00346597).

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

# Appendix

## A   Additional qualitative results

Fig. 9 shows qualitative results on out-of-domain setting, *i.e.*, testing on ScanNet [8] with our method and baseline methods [36, 35, 7] trained on RealEstate10K [52]. We also provide additional qualitative results on in-the-wild images in Fig. 10.

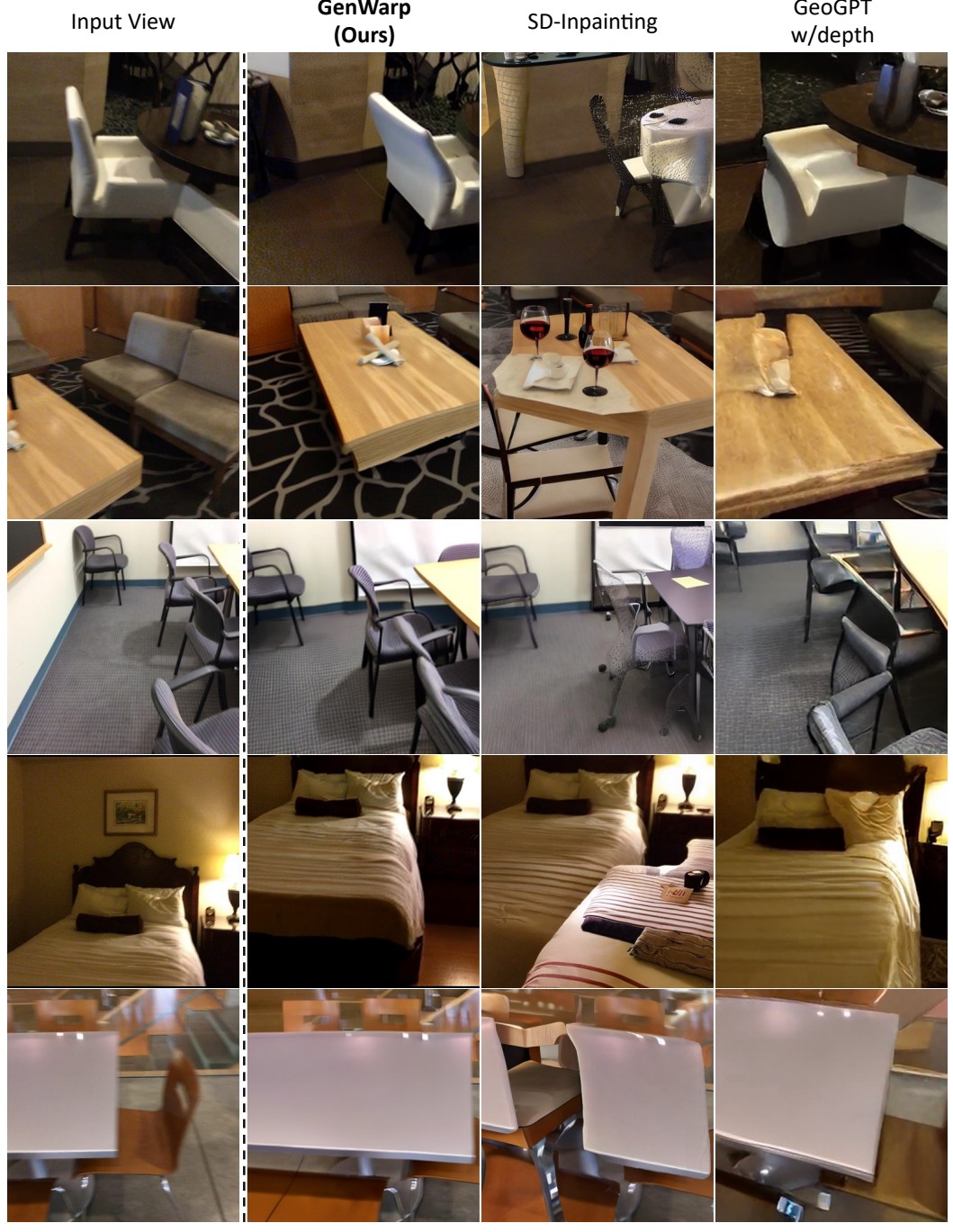

Figure 9: **Extensive qualitative comparisons in out-of-domain setting.** We provide qualitative results of our model trained on RealEstate10K [52], on the external dataset, ScanNet [8].

| Input View | Warped | SD-Inpainting | **GenWarp (Ours)** |
|---|---|---|---|

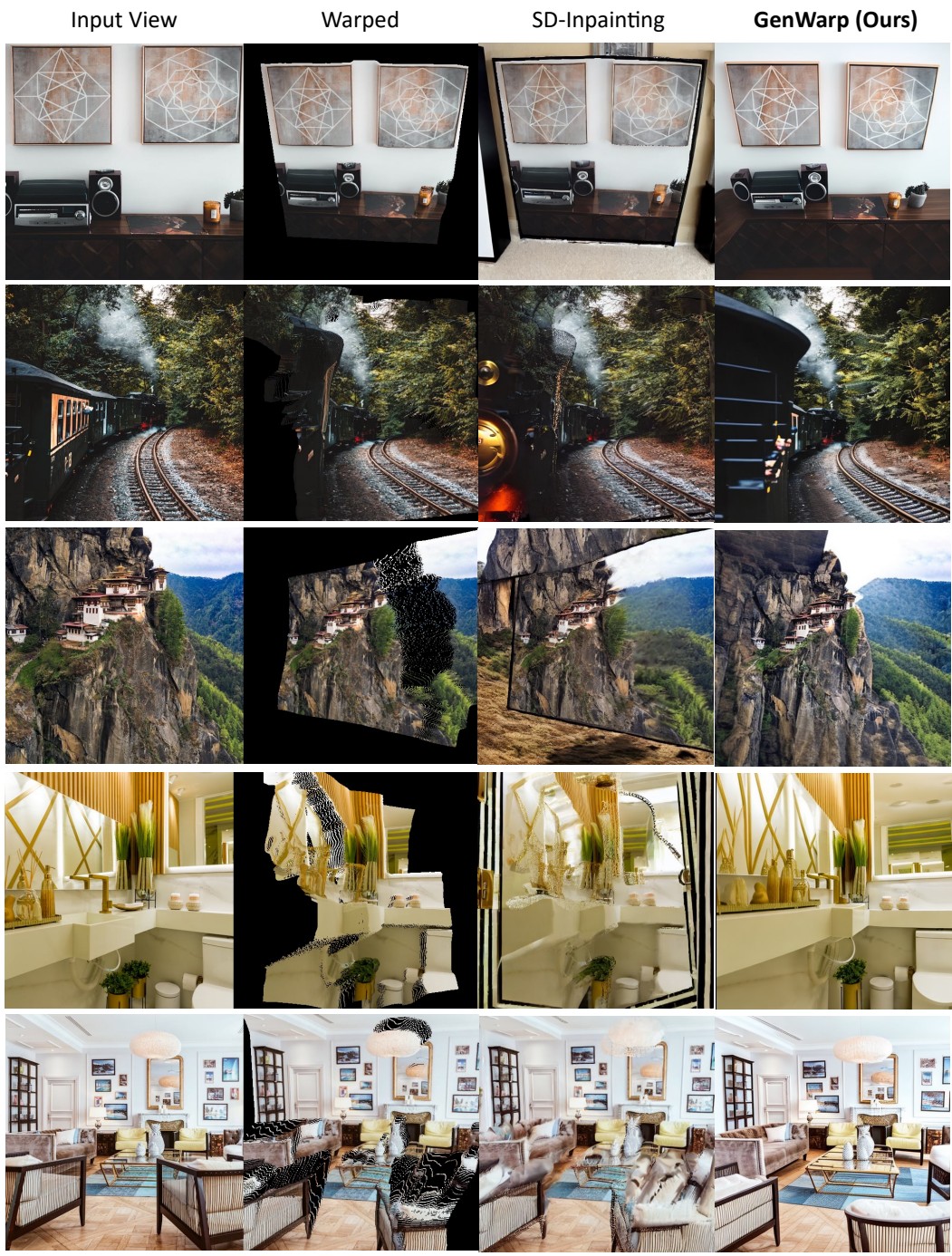

Figure 10: **Extensive qualitative results on in-the-wild images.** We present extensive qualitative results of our method and baseline methods [35, 7] on the in-the-wild images.

# B   Additional implementation details

We initialize our two networks, semantic preserver and diffusion U-net, with Stable Diffusion v1.5 [35], and fine-tune the networks on $2\times$H100 80GB with a batch size of 48 for 2-3 days, at resolutions of $512 \times 384$ and $512 \times 512$. Specifically, we fine-tune the whole parameters of the semantic preserver network and the diffusion U-net in an end-to-end manner. All the hyper-parameters used in our training are kept same as the training of Stable Diffusion 1.5 [2]. In inference, it takes around 2 seconds to generate a novel view with a single H100 80GB.

---

[2]Stable Diffusion v1.5 Model card: https://huggingface.co/runwayml/stable-diffusion-v1-5

**Monocular depth estimation.** We use two external depth estimation networks for all qualitative and quantitative results: ZoeDepth [2] and DUSt3R [46]. DUSt3R is a model that predicts pointmaps given two images as a pair. We use the z-values of these pointmaps in two ways: as a pair depth estimation during training and as a monocular depth estimation during inference (by using the same image for both input views). For quantitative evaluation, we use DUSt3R for depth prediction as the predicted depth maps are passed through the same normalization in the process of DUSt3R with the pseudo depth pairs in training dataset which are estimated using the same network. For qualitative comparisons, we use ZoeDepth to predict metric depth maps. Note that we have used the same estimated depth maps for our method and all the baseline methods [36, 35, 7] which need depth information.

**Reproducing warping-and-inpainting approach with T2I inpainting models.** To implement the warping-and-inpainting strategy using Stable Diffusion Inpainting [36], we follow 'Dream' stage of Lucid-Dreamer [7], which consists of inpainting using the pretrained T2I model [36] after depth-based warping via monocular depth estimation in the official code repository. We observe that directly applying the occlusion mask from depth-based warping to the Stable Diffusion Inpainting model leads to the generation of collapsed images. As suggested in the official code of LucidDreamer, increasing the occlusion mask size for occlusions below a certain threshold effectively prevents this collapse. However, this approach involves a trade-off, as it may further ignore pixels from the source view. Additionally, when using challenging camera trajectories (especially when moving the camera forward), artifacts still occur despite this mask filtering. To address this, we set the minimum occlusion size to $8 \times 8$ and expand smaller occlusions to this size, considering that a resolution of latents in LDM [35] is 8 times lower than that of the images. We use inverse warping for the existing warping-and-inpainting method, which provides natural interpolation and reduces occlusion. In contrast, our method employs forward warping to facilitate the intervention of the generative prior. Fig. 11 shows the difference between forward warping and inverse warping, and the obtained occlusion masks which are used in the subsequent inpainting procedure.

Input View   Forward Warping   Inverse Warping   Occlusion Mask

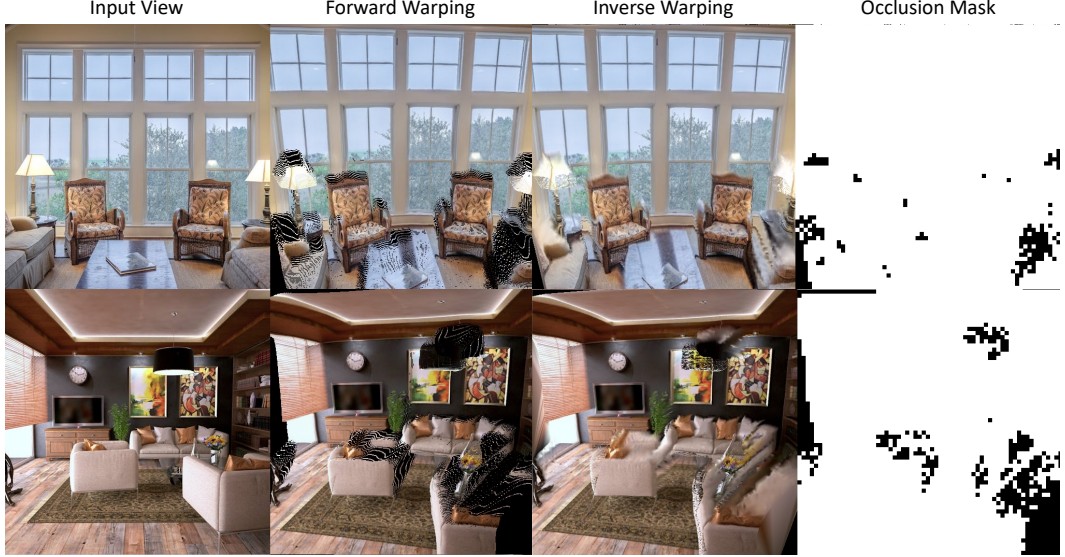

Figure 11: Following LucidDreamer [7], we apply inverse warping and occlusion mask filtering to reproduce the existing warping-and-inpainting approach [7, 31] with Stable Diffusion Inpainting [35].

## C  Additional discussion

**Explicit feature warping vs. implicit warping (ours).** Another straightforward approach for integrating depth-based warping into diffusion models is to warp features within the diffusion model's feature space. We initially tried this diffusion feature warping. Specifically, the input view feature $F_i$ from the semantic preserver network is geometrically warped with the corresponding depth map, and then added to the diffusion U-net's intermediate feature $F_j$ through zero-initialized convolution layers.

This approach shows reasonable performance on multi-view datasets like ScanNet [8], which include ground-truth sensor depth. However, most multi-view datasets [25, 52] are derived from videos and lack dense GT depth. To address this, we use estimated depth maps from DUSt3R [46], as described in Sec. 3.3. Although these pseudo depth pairs are useful, they are not highly accurate. Consequently, we found training a model with explicit feature warping using these pseudo depth pairs leads to instability, as shown in Fig. 12.

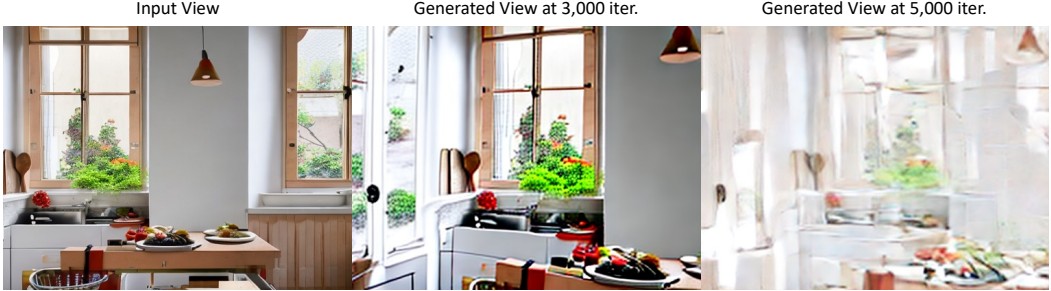

Input View             Generated View at 3,000 iter.           Generated View at 5,000 iter.

Figure 12: Unstable training of an explicit feature warping model using pseudo depth data.

**Additional analysis on viewpoint changes.** We analyze how the performance changes as the ratio of pixels invisible from the input view increases due to viewpoint changes. As shown in Fig. 13 and Fig. 14, our method demonstrates the best performance compared to the other methods even as the invisible ratio increases.

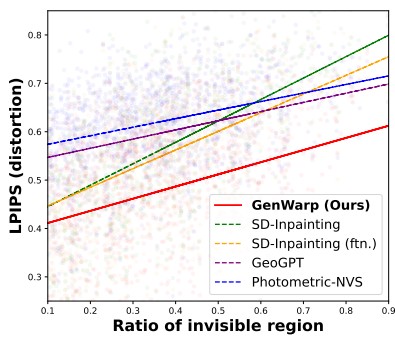

Figure 13: **Comparison of LPIPS with other methods regarding ratio of invisible region.** We measure LPIPS between generated views and GT target views, following GeoGPT [36]'s evaulation protocol.

Figure 14: **Additional ablations on view embedding.** We compare our full pipeline with a variant using camera (Plücker) embeddings and another variant using the camera embeddings and depth information.

**Comparison on reconstruction-based approaches [11, 50, 28].** We provide comparisons with three additional recent/classical methods (Nerdi [11], PixelNeRF [50], vanilla NeRF [28]) in Fig. 15. For Nerdi [11], due to lack of available codes, we brought curated qualitative results on DTU dataset from their paper. Our result shows non-blurry, clear novel view compared to other methods.

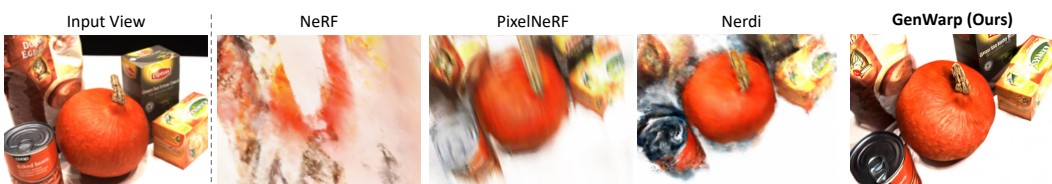

Input View        NeRF          PixelNeRF          Nerdi        **GenWarp (Ours)**

Figure 15: **Comparison with reconstruction-based methods on DTU dataset [18].** Note that our model used here is not trained on DTU dataset.

**Analysis on cross-view attention.** To verify how the cross-view attentions attend to corresponding points, we have used $1,000$ pairs of images to determine (1) how well cross-attention attends to corresponding points, and (2) which is more dominant between self-attention and cross-attention for invisible regions and regions where depth-based correspondence exists. First, Tab. 3 shows the distance between the flow map obtained from depth information and the flow map extracted from the cross-attention. Specifically, we extracted the flow map from the cross-attention layer by argmax operation to see where the model pays the most attention to. It demonstrates that as training progresses, the model learns depth-based matching and warping through the

cross-attention mechanism. On the other hand, the model where the proposed embedding is replaced with the Plücker camera embedding shows relatively worse performance in terms of matching distance.

Secondly, in the Tab. 4, we report which part of the concatenated attention map - the cross-attention part or the self-attention part - is more activated during generation for visible and invisible regions. As exemplified in Fig. 4, it shows the cross-view attention part focuses on regions that can be reliably warped from the input view, while the original self-attention part is more attentive to invisible regions requiring generative priors. Regarding the cross-attention and self-attention for invisible regions, we empirically found that when generating invisible regions, the model also refers to surrounding visible areas through cross-attention, for instance, to generate the invisible left side of a desk, it needs to refer to the visible part of the desk for a plausible novel view.

| Models | Average distance |
|---|---|
| Ours - 2,000 steps | 1.36 |
| Ours - 6,000 steps | 0.97 |
| Ours - 10,000 steps | 0.90 |
| Ours - converged | 0.85 |
| Camera embed. - converged | 0.98 |

Table 3: **Matching distance of models over training steps.**

| Region | Cross-attn. | Self-attn. |
|---|---|---|
| Visible region | 0.756 | 0.244 |
| Invisible region | 0.417 | 0.583 |

Table 4: **Attention distribution in visible/invisible regions.**

**3DGS reconstruction.** Our model can be applied to various downstream tasks. For example, given a single image, our model generates 3-4 novel view images, followed by feeding them into fast 3DGS [20] reconstructors such as InstantSplat [12]. Then we can easily obtain a 3DGS scene in a few seconds. Video examples are shown in the project page.

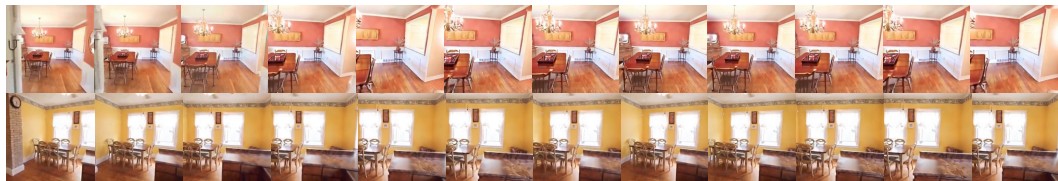

Figure 16: **3DGS reconstruction results.** We show extracted frames in the generated 3DGS videos. Video examples are shown in the project page.

## D    Limitations

Given extremely distant camera viewpoints where depth-based correspondence has no influence, *i.e.*, beyond the unprojected pixels with the depth map, our model struggles with generating novel views. In these cases, instead of generating a novel view in a single step, the approach of sequentially generating novel views conditioned on pre-generated novel views, similar to other single-shot NVS methods [44, 17, 25, 24], should be taken. As with other works [26, 40, 49] that fine-tune pretrained diffusion models, the quality of the dataset used for fine-tuning affects the model's performance. We believe that more high-quality multi-view datasets will maximize the potential of our model.

