# OpenReview forum: "GenWarp: Single Image to Novel Views with Semantic-Preserving Generative Warping"
_NeurIPS.cc/2024/Conference — NeurIPS 2024 poster_

### Official Review · Reviewer_Cjbv · 2024-07-07

**Soundness:** 3
**Presentation:** 4
**Contribution:** 3
**Rating:** 6
**Confidence:** 3

**Summary:**

The authors propose to achieve novel view synthesis from a single image. Contrary to prior work, which warps using a monocular depth estimate and then inpaints, a more flexible architecture is introduced. To do this, a diffusion model is conditioned on warped coordinates, based on the desired pose change and a monocular depth estimate. In addition, the input view (and base coordinate map) is featurized and the diffusion process is allowed to cross attend to these features, which should intuitively allow for attention between corresponding points. The method is then trained end-to-end with a denoising diffusion objective. Experiments show that this approach achieves better FID, PSNR, and LPIPS scores than baselines on RealEstate10k and ScanNet. Ablations justify the choice of warping conditioning.

**Strengths:**

The manuscript is well written and well motivated. The related work section is comprehensive, the methods section is straightforward and easy to understand. And experimental results seem quite strong.

**Weaknesses:**

I think the paper could benefit significantly from apples-to-apples ablations. For example, I am quite interested in a comparison between the proposed method, and the method without any depth information (I think similar to GeoGPT). Or perhaps the proposed method, and the method without the warping.

The authors also claim that the cross-attention “allows the model to inherently decide which regions should rely more on its generative capability and which areas should depend primarily on the information from the input view warping.” However, they provide only a single example of this in figure 4. I think more examples would make for more convincing evidence. I would be interested in a larger scale, more robust study of this hypothesis. For example, this could possibly be automated by computing how closely the cross-attention actually matches warping.

The SD warp baseline could possibly be implemented better. I believe the inpainting mask covers non-warped black pixels, resulting in many unrealistic “black borders” is in Figure 10 and Figure 5. Perhaps a more fair and robust baseline could be achieved by just expanding the inpainting mask a bit.

Text2room by Höllein et al seems relevant. Perhaps the authors would consider citing it.

**Questions:**

From the weaknesses section: could apples-to-apples ablations be conducted on components of the method?

From the weaknesses section: could more evidence for “cross-attention attends to corresponding points” be presented?

One major benefit of the proposed setting, as opposed to warp then inpaint, is that the method should be able to handle view-dependent effects. This may be interesting to investigate.

**Limitations:**

Limitations and societal impacts are included, and adequately addressed, but are not in the main body of the paper.

---

> ### Author Rebuttal · Authors · 2024-08-07
>
> We thank your thoughtful review and suggestions! If any of our responses do not adequately address your concerns, please let us know and we will get back to you as soon as possible.
>
> ---
>
> ### Q: More evidence for “cross-attention attends to corresponding points” be presented?
>
> Thank you for the interesting question!  To verify this, we have used 1,000 pairs of images to determine (1) how well cross-attention attends to corresponding points, and (2) which is more dominant between self-attention and cross-attention for invisible regions and regions where depth-based correspondence exists.
>
> First, the table below shows the distance between the flow map obtained from depth information and the flow map extracted from the cross-attention. Specifically, we extracted the flow map from the cross-attention layer by argmax operation to see where the model pays the most attention to. It demonstrates that as training progresses, the model learns depth-based matching and warping through the cross-attention mechanism. On the other hand, the model where the proposed embedding is replaced with the Plücker camera embedding shows relatively worse performance in terms of matching distance.
>
> | Models | Average distance |
> | --- | --- |
> | Ours - 2,000 steps | 1.36 |
> | Ours - 6,000 steps | 0.97 |
> | Ours - 10,000 steps | 0.90 |
> | Ours - converged | 0.85 |
> | Camera embed.  - converged | 0.98 |
>
> Secondly, in the table below, we report which part of the concatenated attention map - the cross-attention part or the self-attention part - is more activated during generation for visible and invisible regions. As exemplified in Figure 4 of the main paper, it shows the cross-view attention part focuses on regions that can be reliably warped from the input view, while the original self-attention part is more attentive to invisible regions requiring generative priors.
>
> |  | Cross-attn. | Self-attn. |
> | --- | --- | --- |
> | Visible region | 0.756 | 0.244 |
> | Invisible region | 0.417 | 0.583 |
>
> Regarding the cross-attention and self-attention for invisible regions, we empirically found that when generating invisible regions, the model also refers to surrounding visible areas through cross-attention, for instance, to generate the invisible left side of a desk, it needs to refer to the visible part of the desk for a plausible novel view.  We will add this analysis.
>
> ---
>
> ### Q: Apples-to-apples ablations.
>
> Thank you for the suggestion. We report the additional ablation results for the two cases suggested by the reviewer below.
>
> 1. **The method without any depth information**: We have trained our model without depth information and warping process, in which we guided the model with the target camera viewpoint using dense camera embedding, i.e., Plücker embedding.
> 2. **The method without the warping**: We have trained the model with depth information of the input view and the camera embedding, but without the warping process.
>
> In Figure B of the global response PDF, we report a performance comparison between these two cases and our full pipeline. Specifically, we measure LPIPS of each baseline with respect to the ratio of invisible regions in the target camera viewpoint, i.e., the difficulty of generating target views.
>
> It shows that performance improves in the following order, from best to worst: our full model involving the warping process, the model with both camera and depth information, and the model with camera information only. We appreciate this suggestion and will add it to the camera ready.
>
> ---
>
> ### Q: Expanding inpainting mask a bit for SD-inpainting baseline.
>
> Thank you for pointing this out. In the paper, we followed the masking technique of LucidDreamer[7], applying an 8x8 filter to the mask obtained from the warping process, by expanding the mask up to 8x8 size for mask pixels smaller than 8x8.
>
> Actually, at the time of paper submission, we experimented with several mask filter sizes for fair evaluation, and the 8x8 filter achieved the best quantitative results. The table below shows the quantitative results when the mask filter size is increased by 50%.
>
> | Filter size | FID ↓ | PSNR↑ |
> | --- | --- | --- |
> | 8x8 (paper) | 44.13 | 12.98 |
> | 12x12 | 44.19 | 12.88 |
>
> There is a trade-off in mask filter size for the warping-then-inpainting approach — if the filter size is large, it may further ignore pixels from the source view, while conversely, as the reviewer mentioned, artifacts may persist. For details on this, please refer to Figure 11 and L459-L472 in the Appendix.
>
> We thank the reviewer for this point. We will include this discussion and improve the qualitative results of the baseline.
>
> ---
>
> ### Q: Is that the method should be able to handle view-dependent effects? This may be interesting to investigate.
>
> This is an exciting experiment that we have not investigated! Intuitively, our implicit warping, trained on the multi-view datasets where view-dependent effects exist, should better capture these effects compared to explicit geometric warping. These effects could be measured in datasets containing glossy objects, such as Shiny Blender dataset [A]. We will further investigate this and report it in the camera ready as we cannot report it here due to time limitations during the rebuttal phase.
>
> ---
>
> ### Q: Citing Text2Room.
>
> Thank you for the feedback. We will add the Text2Room citation in the camera-ready.
>
> ---
>
> [A] Ref-nerf: Structured view-dependent appearance for neural radiance fields. CVPR. 2022.

---

> > ### Comment · Reviewer_Cjbv · 2024-08-13
> >
> > I thank the authors for answering my questions and providing additional results. The rebuttal addresses my concerns, and I would like to keep my review as a weak accept. I have also read the other reviews and rebuttals, and want to add that I do not think 3Fuk's concerns are significant enough to merit a "borderline reject." Even if they were, the author's rebuttal to the reviews seems very reasonable to me, and I believe 3Fuk's rating should be higher.

---

### Official Review · Reviewer_vdto · 2024-07-07

**Soundness:** 3
**Presentation:** 3
**Contribution:** 3
**Rating:** 6
**Confidence:** 4

**Summary:**

This paper proposes a semantic-preserving generative warping framework to generate high-quality novel views from a single image, which mainly consists of two components:
- Condition the novel view synthesis on the warped 2D coords embedding from the estimated depth map.
- Augmenting cross-view attention with self-attention.

The proposed method eliminates the artifacts caused by error depths in the warping-and-inpainting pipeline and integrates semantic features from source views, preserving semantic details in generation.

**Strengths:**

-	Compared to existing warping-and-inpainting methods that condition on explicit depth warping, the proposed warped 2D coords embedding forms the correspondence between reference view and target view implicitly, which helps the network to be more robust to the noise in the estimated depth map without losing sematic details.

-	The paper is well-written, being clear and easy to follow. The technical limitations are also discussed in detail in the Sup.

**Weaknesses:**

-	The proposed embedding based on depth warping could not benefit the synthesize process when there are large camera movements or occlusions between the input view and the target view.


-	Although the proposed depth-warping embedding somehow reduces the influence of the noise in the estimated depth map, compared to explicit pixel warping, the implicit condition of depth embedding also lower the local preservation ability of the network when synthesizing novel views. Is the model capable of synthesizing consistent novel views which could be used to reconstruct a 3D scene ?


-	The comparison baseline is limited. Plucker embedding[1] is also a dense embedding capable of providing local correspondence, which supports large camera movement and occlusions.  The difference between such dense embedding with the proposed depth warping embedding should be further discussed.


[1] SPAD : Spatially Aware Multiview Diffusers(CVPR2024)[https://arxiv.org/abs/2402.05235]

**Questions:**

As listed in the Weakness.

**Limitations:**

No specific limitation and negative societal impact need to be addressed.

---

> ### Author Rebuttal · Authors · 2024-08-07
>
> We thank your thoughtful review and suggestions! We give a detailed response to your comments below. If any of our responses do not adequately address your concerns, please let us know and we will get back to you as soon as possible.
>
> ---
>
> ### Q: The proposed embedding could not benefit the synthesize process when there are occlusions between the input view and the target view.
>
> Thank you for pointing this out. As the reviewer noticed, the proposed embedding was designed to enhance implicit warping for co-visible regions between input and target views, rather than specifically addressing occlusion. As shown in Figure 4 of the main paper, where the self-attention part of the diffusion model is more attentive to occluded regions, these occluded areas are generated through the generative prior of the diffusion model while referencing co-visible parts.
>
> In the rebuttal, to further validate our model’s performance regarding occlusions,  we calculate the ratio of invisible regions in the warping and analyze how performance changes as this ratio increases. Figure A in the global response PDF shows that our method, along with the proposed embedding, demonstrates **better LPIPS values compared to other methods even when the ratio of invisible regions is high**, showing a similar trend in Figure 8 of the main paper.
>
> In the case of extremely distant viewpoints where depth-based correspondence does not exist, our method, like any other depth warping-based NVS methods[7,31,36], struggles to generate a novel view in a single generation step. We would like to note that in such cases, multi-step progressive generation by re-conditioning on previously generated novel views can be achieved, as similar existing methods [21, 39] have shown (L489).  As exemplified in Figure 6 of the main paper, our model also demonstrates robust performance in consistent view generation.
>
> ---
>
> ### Q: Comparison with Plücker embedding.
>
> We appreciate your feedback. **The comparison between Plücker embedding and our proposed embedding is presented in Table 2 of the main paper**, where Camera embedding [37] refers to the Plücker embedding (L282). We will clarify this point in the camera-ready.
>
> For further comparison, we additionally report the performance comparison of Plücker embedding and our embedding with respect to the ratio of invisible regions. Figure B in the global response PDF shows that the proposed embedding demonstrates better performance consistently as long as there is at least a small overlap between the input view and the target view.
>
>  For the reason why the warped coordinate embedding performs better than the Plücker embedding in our setting, we speculate that the model with the proposed embedding could benefit from the inductive bias that MDE depth and its warping process provide. In other words, Plücker embedding has inherent ambiguity relatively, while the warped coordinate embedding provides a direct warping hint. In our opinion, when fine-tuning with the multi-view training datasets that have relatively less diverse data than other types of datasets, our embedding with this inductive bias can be more efficient.
>
> Thank you for this feedback and we will continue to investigate this!
>
> ---
>
> ### Q: Is the model capable of synthesizing consistent novel views which could be used to reconstruct a 3D scene?
>
> Thank you for the interesting question. To verify this, we reconstructed 3DGS [A] with novel views generated from our model and rendered a video with a camera trajectory that interpolates the given camera viewpoints. We report the video frames in Figure D of the global response PDF, as we are unable to upload the video in the rebuttal. It demonstrates that the 3D scene converges well without being hindered by artifacts.
>
> ---
>
> [A] 3D Gaussian Splatting for Real-Time Radiance Field Rendering. SIGGRAPH 2023.

---

### Official Review · Reviewer_3Fuk · 2024-07-11

**Soundness:** 3
**Presentation:** 2
**Contribution:** 3
**Rating:** 4
**Confidence:** 3

**Summary:**

The paper presents a novel framework called GenWarp, which aims to generate new views from a single input image while preserving the semantic content of the original view. This is achieved by leveraging a generative process that incorporates both self-attention and cross-view attention mechanisms conditioned on warping signals. The proposed approach demonstrates superior performance compared to existing methods, especially in scenarios with challenging viewpoint changes, and exhibits good generalization to out-of-domain images.

**Strengths:**

1.	The paper introduces an approach for single-shot novel view synthesis by combining self-attention and cross-view attention mechanisms to preserve semantic details.
2.	The proposed method outperforms existing techniques in generating high-quality novel views, particularly for challenging viewpoint changes.
3.	GenWarp shows generalization capabilities, performing well on out-of-domain images, which indicates its applicability to various scenarios.

**Weaknesses:**

1.	The paper could further highlight its novelty by providing a more comprehensive comparison with a wider range of state-of-the-art methods, including both classical and recent approaches. Additionally, the authors should discuss any potential limitations of their approach in terms of scalability or adaptability to different types of scenes.
2.	The methodology section could benefit from additional diagrams and flowcharts that illustrate the workflow and attention mechanisms in more detail. Including intermediate results and step-by-step visualizations would help readers better understand the progression from the input image to the generated novel view.
3.	While the paper provides detailed instructions for reproducing the experiments, the code and data are not made publicly available at the time of submission, which can hinder reproducibility efforts.
4.	The performance of GenWarp is highly dependent on the quality of the datasets used for fine-tuning, which could limit its effectiveness if high-quality multi-view datasets are not available.
5.	The method struggles with generating novel views when the camera viewpoints are extremely distant, indicating a limitation in handling very large viewpoint changes.
6.	The reference list could be updated to include more recent advancements in the field, particularly those published in the last year. Additionally, a more detailed comparative analysis of the strengths and weaknesses of related methods would be beneficial.
7. what is the advantage of diffusion-model-based pipeline over 3D Gaussian[A,B] or NeRF[C,D] based pipeline, can the author discuss more about existing methods?

[A] Yu, Zehao, Anpei Chen, Binbin Huang, Torsten Sattler, and Andreas Geiger. "Mip-splatting: Alias-free 3d gaussian splatting." In Proceedings of the IEEE/CVF Conference on Computer Vision and Pattern Recognition, pp. 19447-19456. 2024.
[B] Yan, Zhiwen, Weng Fei Low, Yu Chen, and Gim Hee Lee. "Multi-scale 3d gaussian splatting for anti-aliased rendering." In Proceedings of the IEEE/CVF Conference on Computer Vision and Pattern Recognition, pp. 20923-20931. 2024.
[C] Deng, Congyue, Chiyu Jiang, Charles R. Qi, Xinchen Yan, Yin Zhou, Leonidas Guibas, and Dragomir Anguelov. "Nerdi: Single-view nerf synthesis with language-guided diffusion as general image priors." In Proceedings of the IEEE/CVF conference on computer vision and pattern recognition, pp. 20637-20647. 2023.
[D] Yang, Yifan, Shuhai Zhang, Zixiong Huang, Yubing Zhang, and Mingkui Tan. "Cross-ray neural radiance fields for novel-view synthesis from unconstrained image collections." In Proceedings of the IEEE/CVF International Conference on Computer Vision, pp. 15901-15911. 2023.

**Questions:**

Please refer to the weakness section.

**Limitations:**

Yes.

---

> ### Author Rebuttal · Authors · 2024-08-07
>
> We thank your thoughtful review and suggestions! If any of our responses do not adequately address your concerns, please let us know and we will get back to you.
>
> ---
>
> ### Q: Handling extremely distant viewpoint changes.
>
> Thank you for pointing this out. Our key contribution is effectively using estimated noisy depth signals in NVS diffusion models. While addressing depth-based correspondence for *extremly* distant viewpoints is a common issue for depth-based methods and beyond our scope, our approach outperforms others by implicitly using depth warping signals, even with significant viewpoint changes, as shown in Figure 8 of the main paper.
>
> To further demonstrate this, we analyze how the performance changes as the ratio of pixels invisible from the input view increases due to viewpoint changes. **As shown in Figure A of the global response PDF, our method demonstrates the best performance compared to the other methods even as the invisible ratio increases.**
>
> Finally, we'd like to note that even for extreme viewpoints where depth-based correspondence doesn't exist, progressive generation by re-conditioning on previously generated novel views can be achieved, as similar existing methods[21, 39] have shown. As exemplified in Figure 6 of the main paper, our model also demonstrates robust performance in consistent view generation.
>
> We will faithfully reflect this discussion in the camera-ready version.
>
> ---
>
> ### Q: What is advantage of diffusion-based pipleline over 3DGS/NeRF-based pipleline?
>
> Conventional 3DGS/NeRF pipelines [A,B,D] aim to perform 3D reconstruction using numerous input views, synthesizing novel views through interpolation between input views. However, these methods struggle to synthesize novel views in few-shot scenarios. In such cases, synthesizing a novel view is closer to generation problem than reconstruction problem.
>
> Generalizable NeRF/3DGS pipelines [E,F] learn scene priors through training. In few-shot scenarios, these methods show improved reconstruction performance. However, these works do not explicitly consider generative modeling. Although showing superior reconstruction performance, they show limited performance in synthesizing large unseen areas in novel views, e.g., extrapolation.
>
> GenWarp and other recent methods [7,15,22,33,C] using diffusion models formulate single-shot novel view synthesis as a conditional generation problem, rather than a reconstruction-based approach. Consequently, they show superior performance with extremely limited input views, e.g.,  single-shot scenarios.
>
> ---
>
> ### Q: More comprehensive comparison.
>
> As suggested by the reviewer, we provide comparisons with three additional recent/classical methods (Nerdi [D], PixelNeRF [E], vanilla NeRF) in Figure C of the global response PDF.  For Nerdi [D], due to lack of available codes, we brought curated qualitative results on DTU dataset from their paper. Our result shows **non-blurry, clear novel view compared to other methods**. As their methods are optimization-based and take several hours per scene, we will thoroughly include quantitative comparisons that cannot be done during the rebuttal phase in the camera ready.
>
> Additionally, for comparison with recent methods, we would like to note that we have included a comparison with a warping-then-inpainting strategy with inpainting models [30] which is commonly adopted in recent state-of-the-art pipelines [7,26,36] for single-shot 3D generation.
>
> > Potential limitations of others in terms of scalability or adaptability?
> >
>
> For other NVS generative models [15,31], when these methods are evaluated on different datasets, i.e. in out-of-domain scenarios, they show decreased performance, as shown in Table 1 of the main paper. We speculate that it is because a single dataset usually consists of similar scenes, so the models struggle with different types of scenes unseen during training.
>
> Other approaches [7,26,36] that use warping-then-inpainting with pretrained T2I diffusion models [30], maintain good scalability as they directly utilize the large-scale T2I models without fine-tuning. However, they show unstable results, especially when the target camera viewpoint is far (L106, L49), due to warping errors caused by noisy depths, exemplified in Figure 2 of the main paper. We address these limitations by combining the best of both worlds — GenWarp inherits the generalization capabilities of T2I models while refining the noisy depth-warping artifacts.
>
> ---
>
> ### Q: Dependent on the quality of datasets.
>
> Indeed, our model’s performance is dependent on dataset quality due to its learning-based nature.
>
> However, our model inherits the generalization capabilities of T2I models, which are trained on large, high-quality image corpora. Furthermore, our generative warping approach introduces an inductive bias coming from MDE depth and its warping process, enabling efficient fine-tuning with the training datasets. As a result, it shows superior performance on the same training dataset, as demonstrated by the out-of-domain performance in Table 1 of the main paper.
>
> ---
>
> ### Q: Additional diagram.
>
> Thank you for the feedback. We provide a detailed diagram and intermediate results in Figure E of the global response PDF, which will be included in the camera ready.
>
> ---
>
> ### Q: While the paper provides detailed instructions for reproduction, code and data are not made publicly available.
>
> Thank you for recognizing our effort for reproducibility. We will make sure to release all the code and data in the camera ready.
>
> ---
>
> ### Q: Reference list update.
>
> We thank you for the feedback and will thoroughly update our reference list with the papers [A,B,C,D], as well as recent/concurrent papers. We will also supplement our Related Work section accordingly.
>
> ---
>
> [E] pixelnerf: Neural radiance fields from one or few images. CVPR. 2021
>
> [F] pixelsplat: 3d gaussian splats from image pairs for scalable generalizable 3d reconstruction. CVPR. 2024.

---

### Official Review · Reviewer_2Nrv · 2024-07-12

**Soundness:** 4
**Presentation:** 4
**Contribution:** 3
**Rating:** 6
**Confidence:** 3

**Summary:**

This paper proposes a novel single-shot novel view synthesis framework based on a pretrained T2I diffusion model. Instead of directly warping pixels between input view and novel view,  an implicit approach is proposed to conduct geometry warping operation.  The cross-view attention is used to eliminates the artifacts caused by error depths and integrates semantic features from source views, preserving semantic details in generation. Extensive experiments prove the efffectiveness of the proposed method.

**Strengths:**

1. The implicit geometry warping approach is effective in addressing ill-warped problem and missing original semantic problem encountered in explicit warpping methods.
2. The cross-view attention can provide more useful information for novel view generation.
3. The qualitative and quantatitive experiments on RealEstate10K , ScanNet, and in-the-wild images show the proposed method outperform SOTA methods in both in-domain and out-of-domain scenario.

**Weaknesses:**

1. Although the cross-view attention strategy is effective to fuse features, it may be not novel enough. There are many similar operation in video generation area.
2. The detail of finetuning the T2I model is not clear.

**Questions:**

N/A

**Limitations:**

This method is restricted to depth-based correspondence of two views which limit its application in some scenes where depth correspondence between two views is missing.

---

> ### Author Rebuttal · Authors · 2024-08-07
>
> We thank your thoughtful review and suggestions! We give a detailed response to your comments below. If any of our responses do not adequately address your concerns, please let us know and we will get back to you as soon as possible.
>
> ---
>
> ### Q. Novelty regarding cross-view attention.
>
> Thank you for the feedback. We agree that cross-view attention itself is not newly proposed in our paper — it has been used in the 3D/video generation area.
>
> However, we would like to emphasize that **the novelty of our paper lies not so much in the use of cross-view attention itself, but rather in drawing inspiration from the intuitive connection between cross-view attention and warping operations.**
>
> This led us **guide the attention modules within the diffusion model to emulate geometric warping**. As Table 2 in the main paper demonstrates that naively using cross-view attention yields limited performance, we have proposed to use warped coordinate grids as positional embeddings to support our generative warping.
>
> Our other strategy is aggregating the cross-view attention with the self-attention in the diffusion model at once, instead of inserting new cross-view attention layers. With this strategy, we found that the self-attention effectively finds where to refine, compensating for warping errors coming from noisy MDE depth, exemplified in Fig.4 of the main paper.
>
> By doing so, we have overcome the performance constraints inherent in existing two-step approaches that rely on the warping-then-generation paradigm. We believe these findings possess robust merits that contribute to this field.
>
> ---
>
> ### Q: The detail of finetuning the T2I model.
>
> Thank you for pointing this out. As described in L445-449, we initialize our two networks, semantic preserver and diffusion U-net, with Stable Diffusion v1.5 [30], fine-tune the networks on 2×H100 80GB with a batch size of 48, at resolutions of 512 × 384 and 512 × 512. We used a learning rate of 1.0e-5, which is the same as the value used when training Stable Diffusion, and we kept all other hyperparameters at the same values used in Stable Diffusion.
>
> Additionally, the coordinate embeddings are passed through 3 convolutional layers and added to the input of the diffusion U-Net and the semantic preserver network. All parameters of our model are trained in an end-to-end manner through the diffusion loss shown in Equation 6 of the main paper. We will clarify these points in the camera-ready version.
>
> If we have not addressed any unclear aspects, please let us know, and we will reply accordingly.
>
> ---
>
> ### Limitation section.
>
> As the reviewer mentioned, we discuss the viewpoint limitation due to depth-based correspondence in Limitation section — as this is not the goal of our paper, our method does not explicitly consider this common issue in depth-based methods.
>
> However, we would like to remind you that (1) as shown in Figure 8 of the main paper, our method demonstrates superior performance compared to other methods even when viewpoint changes are large, as long as there is at least a small overlap between the views, (2) even for extremely distant viewpoints, multi-step progressive generation by re-conditioning on previously generated novel views can be achieved (L487-L489), exemplified in Figure 6 in the main paper.

---

### Author Rebuttal · Authors · 2024-08-07

# General Response

We would like to first thank the reviewers for the helpful suggestions and constructive reviews. We are greatly encouraged by their positive assessment regarding soundness (1 excellent, 3 good), contribution (4 good), and presentation (2 excellent, 1 good) of our work.

They acknowledge that our generative geometric warping is **effective in addressing ill-warped problems** (2Nrv), **preserving semantic details** (2Nrv,3Fuk), exhibiting **generalization capabilities** (3Fuk), our warped 2D coords embedding helps the network to be **more robust to the noisy predicted depth maps** (vdto), and our manuscript is **well written** (Cjbv, vdto) and **well motivated** (Cjbv).

We also thank the reviewers for recognizing that our **experimental results are strong** (Cjbv), outperforming existing methods in **both in-domain and out-of-domain scenarios** (2Nrv) under **challenging viewpoint changes** (3Fuk).

---

In the rebuttal, we have conducted the following additional experiments to address the reviewers' questions and suggestions:

- We provide further analysis on performance changes as the ratio of invisible regions varies. (Figure A in the attached PDF file)
- We conduct additional ablation studies on camera (Plücker) embedding and depth embedding. (Figure B in the attached PDF file)
- To verify whether cross-view attention mimics depth-based warping, we analyze the distance between GT flow and flow extracted from the cross-view attention. (First table in the response to Reviewer Cjbv)
- We analyze the impact of cross-view attention and self-attention on visible and invisible regions. (Second table in the response to Reviewer Cjbv)
- For extensive comparison, we provide a comparison with existing NeRF-based pipelines. (Figure C in the attached PDF file)
- We provide results of 3D scene (3DGS) reconstruction with novel views generated by our method. (Figure D in the attached PDF file)
- We provide intermediate generation results. (Figure E in the attached PDF file)

---

### Decision · Program_Chairs · 2024-09-25

**Decision:**

Accept (poster)

**Comment:**

The paper addresses novel view synthesis from a single image. Contrary to prior work, which warps using a monocular depth estimate and then inpaints, the diffusion model is conditioned on warped coordinates, making the approach more robust noisy depth estimates.

Overall, the reviewers are positive about the paper, mentioning strong results and improvements over baselines in challenging settings. The main concern was to add comparisons to more baselines which the authors addressed in their rebuttal.

I reccomend acceptance.